# Oestrogen and Vibration Improve Intervertebral Disc Cell Viability and Decrease Catabolism in Bovine Organ Cultures

**DOI:** 10.3390/ijms24076143

**Published:** 2023-03-24

**Authors:** Franziska Widmayer, Cornelia Neidlinger-Wilke, Fiona Witz, Jan U. Jansen, Anita Ignatius, Melanie Haffner-Luntzer, Graciosa Q. Teixeira

**Affiliations:** Institute of Orthopaedic Research and Biomechanics, University of Ulm, 89081 Ulm, Germany

**Keywords:** intervertebral disc, oestrogen, 17β-oestradiol, vibration, organ culture, matrix proteins, matrix metalloproteinases, anabolism, catabolism

## Abstract

Postmenopausal women are at an increased risk for intervertebral disc degeneration, possibly due to the decrease in oestrogen levels. Low-magnitude, high-frequency vibration (LMHFV) is applied as a therapeutic approach for postmenopausal osteoporosis; however, less is known regarding possible effects on the intervertebral disc (IVD) and whether these may be oestrogen-dependent. The present study investigated the effect of 17β-oestradiol (E2) and LMHFV in an IVD organ culture model. Bovine IVDs (*n* = 6 IVDs/group) were treated with either (i) E2, (ii) LMHFV or (iii) the combination of E2 + LMHFV for 2 or 14 days. Minor changes in gene expression, cellularity and matrix metabolism were observed after E2 treatment, except for a significant increase in matrix metalloproteinase (MMP)-3 and interleukin (IL)-6 production. Interestingly, LMHFV alone induced cell loss and increased IL-6 production compared to the control. The combination of E2 + LMHFV induced a protective effect against cell loss and decreased IL-6 production compared to the LMHFV group. This indicates possible benefits of oestrogen therapy for the IVDs of postmenopausal women undergoing LMHFV exercises.

## 1. Introduction

Back pain is a leading global cause of disability [1], with a considerable loss of productivity and high healthcare costs [2]. It is more prevalent in adults aged 40–80, particularly in females [3]. Intervertebral disc (IVD) degeneration (IVDD) is perceived as the major cause of back pain [4]. Female sex hormones have been described to play an important role in the aetiology and pathophysiology of several musculoskeletal degenerative diseases. An association between menopause and lumbar IVDD has been previously observed, suggesting oestrogen deficiency to be a risk factor for IVDD [5,6,7,8,9,10]. Accelerated IVDD has been observed in women during the first 15 years post-menopause due to relative oestrogen deficiency, increased prevalence of spondylolisthesis and facet joint osteoarthritis [5,7,10,11]. Nevertheless, women are diagnosed and treated similarly to men, and possible differences in how the two sexes experience IVDD have been little explored.

Several studies demonstrated positive effects of oestrogen-related signalling on IVD metabolism. In two-dimensional (2D) cultures, 17β-oestradiol (E2) was shown to enhance human IVD cell proliferation [12,13], and to exert anti-apoptotic effects in human cartilaginous endplate cells [14] and rat nucleus pulposus (NP) [15,16]. In ovariectomised female rats with IVDD, a 28-day treatment with E2 downregulated matrix metalloproteinase (*MMP*)*3* and *MMP13* expression and up-regulated type II collagen (*COL2A1*) expression [17]. This indicated that E2 intradiscal injection may modulate IVD extracellular matrix (ECM) production towards anabolism; however, it is not known whether E2-induced changes at the gene expression level can be translated into an increase in IVD matrix production. In bone tissue, osteoanabolic effects were shown to be mediated via enhanced chondroitin sulphate production [18], an important sulphated glycosaminoglycan (sGAG) in IVD metabolism. Depletion of sGAG within the IVD during degeneration results in a decrease in tissue hydration, cell apoptosis, loss of nerve growth inhibition and ultimately, loss of disc function [19].

Studies in Chinese [20] and Egyptian [21] populations suggested that whole-body vibration, among other factors, may be associated with IVDD. However, whole-body vibration is a popular fitness trend that is described to increase muscle mass, induce weight loss and reduce joint pain, and is one of the types of physical activity recommended for the treatment of postmenopausal osteoporosis [22]. In mice, it was shown that the impact of low-magnitude high-frequency vibration (LMHFV) under conditions that mimic those used clinically in humans (0.3 g peak-to-peak acceleration/45 Hz, 20 min/day, 5 days/week) on fracture healing was highly influenced by oestrogen levels, and that this mechanism was dependent on oestrogen receptor-α signalling and cytoskeletal remodelling [23,24]. Repeated exposure to whole-body LMHFV (0.3 g peak-to-peak acceleration/45 Hz, 20 min/day, 5 days/week) was also shown to induce meniscal tears and articular cartilage damage after 4 weeks of stimulus [25]. Progressive IVDD associated with increased expression of pro-inflammatory interleukin (IL)-1β and multiple matrix degrading enzymes, including MMP-3 and MMP-13, was also observed after 8 weeks of LMHFV [26]. Interestingly, IVD matrix components aggrecan and COL2 were also upregulated by vibration [26]. Overall, the data from the different publications indicate that vibration may play a contradictory role in IVD ECM metabolism.

On the basis of the discussed literature, we hypothesise that the effects of E2 and LMHFV might interact with each other in the IVD. We assume that (i) E2 intradiscal injection may modulate IVD ECM production towards anabolism, as suggested by Liu et al. (2018) [17], and (ii) LMHFV in combination with E2 may amplify an anabolic response by IVD cells.

## 2. Results

To investigate the proposed hypotheses, bovine IVD organ cultures were established (Figure 1A). On day 6 of culture, the IVDs were treated with (i) 17β-oestradiol (oestrogen, E2), (ii) LMHFV or (iii) both in combination (E2 + LMHFV). E2 was added to the medium at a concentration of 100 nM [27] and was added to the medium at every medium exchange. The groups treated with LMHFV were placed on a custom-made vibration platform (Figure 1B) and exposed to a frequency of 45 Hz and peak-to-peak acceleration of 0.3 g for 20 min per day, 5 days per week [28]. Samples were collected after 2 or 14 days of treatment (days 8 and 21 of organ culture, respectively) (Figure 1C). The study design is illustrated in Figure 1D.

### 2.1. Effect of Oestrogen and Vibration on the Expression of Genes of the Extracellular Matrix and Its Metabolism

After 2 days of incubation of bovine IVDs under basal control conditions or with E2, LMHFV or both E2 + LMHFV, the gene expression of cell metabolism and apoptosis marker *PT53*, oestrogen receptors *ESR1* and *ESR2,* pro-inflammatory gene prostaglandin-endoperoxide synthase 2 (*PTGS2*) and in annulus fibrosus (AF) and NP cells was assessed by quantitative reverse transcription-polymerase chain reaction (qRT-PCR). Prostaglandin E_2_ (PGE_2_) was quantified in the culture supernatants at days 8 and 21. The expression of *PT53* appeared to be downregulated in AF and NP cells after E2 treatment in comparison to the control group (0.06 ± 0.09- and 0.05 ± 0.03-fold in AF and NP, respectively, *p* = 0.062; Figure 2A). *ESR1* and *ESR2* were detected in AF and NP cells (Figure 2B,C). *ESR1* expression was not altered in AF cells, but downregulated in NP cells after E2 or LMHFV treatments, both when compared to the control (0.13 ± 0.20- and 0.31 ± 0.06-fold for E2 and LMHFV, respectively, *p* < 0.05) and E2 + LMHFV (0.11 ± 1.6-fold for E2, *p* < 0.01, and 0.26 ± 0.56-fold for LMHFV, *p* = 0.067; Figure 2B). LMHFV appeared to downregulate 0.35 ± 0.12-fold (*p* = 0.062) *ESR2* expression in AF cells, while in NP cells it was downregulated by either E2 (0.12 ± 0.08-fold, *p* = 0.062) or LMHFV alone (0.31 ± 0.15-fold, *p* < 0.05; Figure 2C). *PTGS2* was slightly upregulated after the E2 stimulation in AF cells (17 ± 27-fold) and significantly upregulated in NP cells when compared to control conditions (10 ± 12-fold, *p* < 0.05, Figure 2D). Additionally, *PTGS2* in AF cells treated with E2 + LMHFV was upregulated versus the control (36 ± 46-fold, *p* = 0.062) and the LMHFV group (34 ± 1.7-fold, *p* < 0.05). At day 8, PGE_2_ production was significantly higher in E2 (2.7 ± 0.49-fold, *p* < 0.05) and E2 + LMHFV (5.4 ± 0.45-fold, *p* < 0.01) compared to the control, and in E2 + LMHFV compared to LMHFV alone (6.0 ± 0.56-fold, *p* < 0.01, Figure 2E). At day 21, significantly lower PGE_2_ production was observed in E2 (0.26 ± 0.57-fold, *p* < 0.001) and E2 + LMHFV (0.19 ± 0.67-fold, *p* < 0.0001) groups versus day 8, with no differences between the groups at the later timepoint.

The expression of ECM components collagen type I alpha 1 chain (*COL1A1*) and *COL2A1*, as well as of matrix-degrading enzymes (*MMP1*, *MMP3* and a disintegrin and metalloproteinase with thrombospondin motifs 4 [*ADAMTS4*]) and their inhibitors (tissue inhibitor of MMPs 1 [*TIMP1*], *TIMP2*) was also investigated by qRT-PCR after 2 days of treatment of bovine IVDs. No effects on the regulation of *COL1A1* or *COL2A1* were observed in AF cells (Figure 3A,B). By contrast, in the NP, LMHFV led to a downregulation of *COL1A1* expression (0.41 ± 0.27-fold), whereas *COL2A1* upregulation (3.4 ± 2.4-fold) was observed when compared to basal control conditions (*p* < 0.05; Figure 3A,B). LMHFV in combination with E2 also led to *COL2A1* upregulation versus the control in the NP (9.0 ± 12-fold, *p* < 0.05; Figure 3B). Aggrecan *(ACAN)* was not altered in the AF, but was slightly upregulated by E2 alone or E2 + LMHFV in the NP (7.5 ± 6.6- and 13 ± 7.1-fold, respectively, *p* = 0.062; Figure 3C). Regarding matrix degradation, no differences were observed in *MMP1* expression (Figure 3D). In comparison to the control group, *MMP3* was slightly downregulated by AF cells stimulated with LMHFV (0.48 ± 0.37-fold, *p* = 0.062) and upregulated by NP cells cultured with E2 medium supplementation (13 ± 10-fold, *p* = 0.062; Figure 3E). Additionally, *ADAMTS4* expression was not altered in the AF, but was significantly downregulated by all treatments in the NP (0.26 ± 0.31-, 0.31 ± 0.14- and 0.14 ± 0.19-fold for E2, LMHFV and E2 + LMHFV, respectively, *p* < 0.05; Figure 3F). A trend could be seen towards *TIMP1* downregulation in NP cells treated with E2 + LMHFV (0.11 ± 0.17-fold, *p* = 0.062; Figure 3G). *TIMP2* was downregulated by LMHFV in the AF (0.26 ± 0.19-fold, *p* < 0.01) and upregulated by E2 in the NP (3.0 ± 1.4-fold, *p* < 0.05) compared to the control, while combined treatment significantly attenuated these effects (*p* < 0.05; Figure 3H).

### 2.2. Effect of Oestrogen and Vibration on Cell Viability, DNA, sGAG and Collagen

After 21 days of culture, including 16 days of stimulation with either E2, LMHFV or E2 + LMHFV, cell apoptosis in the AF and NP was investigated by the terminal deoxynucleotidyl transferase dUTP nick-end labelling (TUNEL) assay (single staining channels are depicted in Appendix A, Figure A2). The amount of DNA was assessed in digested tissues. Cell viability was over 70% in the AF, and over 58% in the NP, and no significant differences were observed between the groups (Figure 4A,B). The DNA content was normalised to the wet weight of each respective tissue sample (Figure 4C). A significant decrease in the DNA was observed in AF and NP tissues treated with LMHFV when compared to both control (0.43 ± 0.54 and 0.21 ± 0.27-fold for AF and NP, respectively, *p* < 0.05) and E2 + LMHFV (0.17 ± 0.63 and 0.10 ± 0.43-fold for AF and NP, respectively, *p* < 0.05).

The amounts of sGAG and collagen were normalised to the DNA content of each tissue digest (Figure 5). Representative images of sGAG and collagen content are depicted in Figure 6A,B, respectively. While no differences were observed between groups in the AF, the amount of sGAG/DNA in the NP was significantly increased after the application of LMHFV (1.9 ± 0.40-fold, *p* < 0.05, Figure 5A). This effect was reversed when E2 was also present in the combined group. Samples treated with LMHFV showed displayed higher collagen/DNA both in the AF (2.3 ± 0.52-fold, *p* < 0.05) and NP (2.0 ± 0.30-fold, *p* < 0.01) regions, whereas the combination of E2 + LMHFV significantly decreased it (0.62 ± 0.60-and 0.27 ± 0.54-fold in the AF and NP, respectively, *p* < 0.05, Figure 5B). Interestingly, when E2 was administered alone, no differences were observed compared to control samples.

When analysing in more detail the distribution of collagen fibres (Figure 6B), the total content was in agreement with the increase of collagen in the LMHFV-treated group detected by biochemical quantification (Figure 5B). The % of collagen detected by picrosirius red staining also showed a significantly higher collagen content in the LMHFV-treated group compared to E2 + LMHFV, particularly in the AF region (1.4 ± 0.22-fold, *p* < 0.05). Regarding the distribution of fibres with different thicknesses, it was observed that orange fibres (relatively thick and mature) were the most abundant in all groups (Figure 6C). Additionally, more orange fibres were detected in the AF of LMHFV-treated IVDs versus E2 + LMHFV (1.7 ± 0.26-fold, *p* < 0.01). Though less than 7% of yellow fibres were detected in all groups, a higher content was observed in the AF of E2 + LMHFV compared to E2 alone (2.1 ± 0.68-fold, *p* = 0.059) and of green fibres compared to LMHFV (4.1 ± 1.2-fold, *p* < 0.05). In the NP, a higher content of yellow and green fibres was observed after LMHFV was applied (3.2 ± 0.98- and 3.3 ± 1.3-fold, respectively, *p* < 0.05).

### 2.3. IL-6 and MMP-3 Production

Immunohistochemical staining for IL-6 (Figure 7A), an inflammation-associated molecule, and MMP-3 (Figure 7B), a matrix-degrading enzyme, was performed on IVD sections collected at day 21 of organ culture after treatment with E2, LMHFV or E2 + LMHFV was started at day 6. Representative images of AF and NP regions are depicted in Figure 7A,B and of negative immunoglobulin G (IgG) control in Appendix A Figure A5. For each experiment, all groups were stained at the same time for comparative purposes. The staining intensity in each treatment group was normalised to the respective untreated control. Spatial distribution investigations depicted stronger staining intensity of both proteins in pericellular regions. Significantly higher IL-6 production was observed in the AF and NP regions of the E2- and LMHFV-treated groups versus the control (about 1.2 ± 0.06-fold, *p* < 0.05; Figure 7C). IVDs treated with E2 + LMHFV also produced more IL-6 in comparison to control samples in the NP region (1.1 ± 0.06-fold, *p* < 0.05), but significantly less than IVDs treated with E2 or LMHFV alone (0.9 ± 0.07-fold, *p* < 0.05).

MMP-3 production was significantly increased in the AF with E2 treatment (1.1 ± 0.03-fold, *p* < 0.01), particularly in the pericellular region and in the translamellar bridging network, but decreased with LMHFV and E2 + LMHFV in combination, when compared to the control group (about 0.93 ± 0.03-fold, *p* < 0.05; Figure 7D). In addition, the combined treatment showed significantly less MMP-3 than did E2 alone (0.89 ± 0.03-fold, *p* < 0.0001). In the NP region, only E2 + LMHFV led to significantly lower MMP-3 production in comparison to control samples (0.95 ± 0.04-fold, *p* < 0.05).

## 3. Discussion

IVDD is characterised by, among other events, an imbalance in ECM metabolism favouring degradation/catabolism [29,30]. While a lack of E2 has been associated with IVDD [5,17,31,32], the influence of LMHFV on IVD homeostasis and degeneration is, to date, not fully understood [25,26,33]. Therefore, the main aim of the present work was to investigate the effects of E2 and LMHFV, alone or in combination, on IVD matrix metabolism, using an organ culture model of male IVDs. At day 8 of organ culture, *ESR1*, but not *ESR2*, was upregulated by AF and NP cells compared to fresh samples (Appendix A, Figure A1), which might have been influenced by the culture conditions [34]. However, we do not expect significant differences between sex regarding ex vivo findings because the physiological influence of hormones such as oestrogen was not present. Additionally, significant cell death and GAG release were observed at day 21, when compared to fresh IVDs (Appendix A, Figure A3 and Figure A4), simulating features of mildly degenerated IVDs. Decrease in cell viability and GAG loss in bovine IVD organ cultures without endplates with time in culture have been previously observed under application of similar static [35] or dynamic [36] loads. Given the difficult access to healthy human IVDs for research, bovine caudal IVDs have been proposed as a suitable biological and biomechanical model to investigate the human lumbar IVD given their similarities in size, cellular composition and metabolic behaviour [37]. Explant cultures present limitations in the simulation of physiologic hydration levels and hydrostatic pressure conditions and lack the interaction with other tissues; however, they are easily available and allow for well-controlled environmental conditions [37,38].

*ESR1* and *ESR2*, classic nuclear receptors of oestrogen signalling, are expressed by human AF [13] and NP cells, which was confirmed in the present work for the bovine cells, and their expression was shown to be significantly decreased in the NP with the aggravation of IVDD both in samples from male and female patients [39,40]. The expression of ERα and ERβ proteins was previously shown to be higher in the NP of degenerated IVDs of males versus females [40]. Nevertheless, as stated above, we do not expect significant differences between sex regarding ex vivo findings. In the present study, a single administration of 100 nM E2 to IVD tissues upregulated *ESR1* and *ESR2* expression in NP cells after 2 days. Previously, human NP cell cultures treated with 10 nM E2 (in combination with 15 ng/mL tumour necrosis factor-α [TNF-α]) for 3 days increased their expression of ERα and ERβ in comparison to the TNF-α treatment alone, but up to similar values as in the control group [41].

On the basis of the current literature [17,31], we hypothesised an anabolic effect of E2 on the IVD potentially mediated via sGAG production [18]. This hypothesis could not be confirmed in the present study. Here, the expression of *PTGS2* and production of PGE_2_, and the expression of *ADAMTS4*, as well as of *TIMP2* (an inhibitor of MMP activity) were downregulated by NP cells after E2 treatment, whereas *COL1A1*, *COL2A1*, *MMP1* and *MMP3* expression was not changed. In addition, immunohistochemical analysis suggested a pro-inflammatory and catabolic effect of E2 through increased IL-6 and MMP-3 production; however, with no significant changes in sGAG or collagen content. No changes in cell viability or cellularity were observed, in contrast to previous work in which human AF cell proliferation was observed in vitro following exposure to 100 nM E2 for 10 days [13]. E2 administration (10 µM, 100 nM and 1 nM for 48 h) was shown to upregulate *SOX9*, *ACAN* and *COL2A1* gene expression by NP cells of male rats cultured in vitro, and to enhance protein deposition of ACAN and COL2, and sGAG in a concentration-dependent manner [31]. The p38 mitogen-activated protein kinase (MAPK) signalling pathway was shown to be involved in this regulatory process [31]. Rat NP cells treated with 1 nM E2 for 24 h were shown to be protected against apoptosis by upregulating α2β1 signalling pathway [15] and COL2, downregulating MMP-3 and MMP-13, and by inhibiting the activation of the nuclear factor-κB (NF-κB) signal pathway, which plays a relevant role in IVDD [16,42]. The E2 effects were also demonstrated to occur via the PI3K/Akt/mTOR pathway, a key regulator of survival during cellular stress [43,44,45]. In vivo, E2 administration (25 μg/kg body weight/day for 28 days) attenuated ovariectomy-induced IVDD in female rats by *MMP3* downregulation and *COL2A1* upregulation [17]. In ovariectomised female mice, E2 treatment (subcutaneous implantation of pellets containing 0.18 mg E2) decreased tumour necrosis factor-α, IL-1β and IL-6 in NP cells at the gene and protein levels after 6 weeks [46]. Interestingly, E2 modulation of these pro-inflammatory cytokines was shown to be mediated by substance P [40,46]. In summary, only a mild effect of E2 could be observed on IVD matrix metabolism up to 14 days after treatment of bovine IVD organ cultures. Although the used E2 concentration was is in the range of published work for 2D cells cultures (between 1 nM and 10 µM) [13,15,31,39], oestrogen bioavailability may be lower in organ culture when compared to isolated cells or endogenous production, and the effect of soluble factors on isolated IVD cells may be inherently different than on three-dimensional organ cultures, as previously shown in other investigations [47]. Nevertheless, the observed differences may be mostly due to the fact that the cells were not challenged with, for instance, pro-apoptotic [15] or pro-inflammatory factors [39,42,44,45], as performed in previous studies. Yet, the aim of this work was to investigate the effect of E2 only under homeostatic/mildly degenerative conditions.

Few studies have addressed the effects of vibration, particularly LMHFV, on the IVD [48]. Interestingly, acute vibration at 15 Hz (0.3 g peak-to-peak acceleration, 30 min) was shown to induce transient expression of anabolic genes including *Acan*, *Col2a1*, biglycan, decorin, and *Sox9*, and suppress expression of *Mmp13* by murine IVDs cultured ex vivo, with the most pronounced changes detected 6 h following vibration [33]. Similar results were detected in mouse IVDs in vivo; however, anabolic effects decreased at frequencies of 45 Hz or greater (up to 90 Hz) [33]. Longer exposure to 45 Hz (0.3 g peak-to-peak acceleration, 30 min/day, 5 days/week, 4 weeks) promoted degenerative changes. *MMP3* and *COL1A1* expression was found to be increased by the mouse IVD cells, as well as the amount of sGAG in the AF [25]. In our experiments, LMHFV applied for 2 days downregulated *COL1A1* expression by NP cells, but increased *COL2A1* expression, typical of NP cells in the native environment, in line with the findings from McCann et al. (2013) for a short exposure time [33]. Although no significant differences in cell apoptosis were detected by the TUNEL assay, the cellular content of the discs after 2 weeks of LMHFV was decreased, also in agreement with previous findings [25]. In addition, lower sGAG content was observed in the NP of LMHFV-treated samples, which may be associated with the fact that sGAG can easily diffuse out of the tissue, whereas collagen cannot. While there was a decreased MMP-3 production, a known activator of collagen degradation [49], an increased percentage of orange “more-mature” collagen fibres was observed in the AF of LMHFV-treated IVDs. This may indicate an impact of LMHFV on collagen glycosylation, which has been previously described to be implicated in fibrillar collagen maturation [50]. In humans, short applications of LMHFV (30 Hz, peak-to-peak acceleration magnitudes of either 0.3 or 0.5 g, 10 min/day) attenuated IVD swelling after 90 days of bed rest [51]. Overall, these findings demonstrate that the effects of LMHFV on IVD metabolism are dependent on the frequency and duration of the stimulus, as well as the timepoint of analysis. While short-term vibration at 45 Hz does not seem to induce strong changes in gene expression [33], repeated exposure may induce deleterious effects on healthy IVDs [25].

Lastly, the effect of the combined E2 + LMHFV treatment on IVDs was, to the best of our knowledge, here investigated for the first time. Our data showed an increase in *COL2A1* expression by NP cells, as well as cell proliferation and decreased MMP-3 production. However, no changes in sGAG were observed and collagen was even decreased. The higher cell content in the IVD after E2 + LMHFV was applied for 14 days indicates that E2 may abrogate the detrimental effects of LMHFV alone, in contrast to results from Haffner-Luntzer et al. (2018), in which LMHFV-induced cell proliferation of osteoblasts was abolished by oestrogen supplementation [23]. Cell proliferation may contribute to the prevention of IVDD. The decrease of collagen suggested a lower production by the cells, which may be compensated at a later timepoint by a higher cellularity of the disc. E2 might have had an effect on LMHFV-activated cell death, for instance, via PI3K/Akt/mTOR pathway, as previously suggested [43,44,45]. A decrease in IL-6 production may have occurred, for example, via modulation of the NF-κB signalling pathway [16,42]. In bone tissue homeostasis, the individual anabolic effects of E2 and LMHFV were stronger than when combined [24]. A positive effect of LMHFV was observed on fracture healing repair in ovariectomized mice (which were not capable to produce oestrogen), whereas in non-ovariectomized healthy mice, LMHFV negatively affected bone repair [23,24]. These effects were shown to be dependent on oestrogen receptor α signalling and cytoskeletal remodelling [23]. The contrast to our study may be attributed to possible differences in the pathomechanisms activated in bone and IVD, though different animal models and experimental setups also may play a role. Although further research will be important to better understand the mechanism of action of E2 and LMHFV on IVD metabolism both in health and disease, the present study suggests a beneficial effect of E2 + LMHFV in healthy/mildly degenerated IVDs, representing a possible therapeutic option to decelerate IVDD development.

In conclusion, this study provides insights into the effect of E2 and LMHFV on IVD organ cultures, either singly or in combination. Overall, little effects were observed at the gene expression level (Figure 8). However, E2 significantly increased the production of IL-6 and MMP-3, involved in IVD matrix catabolism, whereas LMHFV applied alone contributed to a decrease in cellularity and an increase in IL-6, but also in collagen compared to the control. The combination of E2 + LMHFV decreased the production of MMP-3 compared to the control. Additionally, it promoted cell proliferation and decreased IL-6 production compared to the LMHFV group, suggesting possible benefits for IVD homeostasis in postmenopausal women undergoing oestrogen treatment and whole-body vibration. Nevertheless, further research is important to better understand the effect and mechanism of action of E2 and LMHFV on IVDD.

## 4. Materials and Methods

### 4.1. IVD Isolation and Organ Culture Model

Tails from male cattle between the ages of 12 and 24 months (*n* = 12) were obtained from a local slaughterhouse (Ulmer Fleisch, Ulm, Germany)—no ethical approval was required. The tails were processed as previously described [35]; after removing muscles and ligaments, the mid-region of each IVD was isolated from the caudally and cranially adjacent cartilaginous endplates using a custom-built cutting-tool, containing two parallel microtome blades 5 mm apart. Six IVDs per tail (with 5 mm height) were collected. The IVDs were washed in Dulbecco’s phosphate-buffered saline (Gibco, Waltham, MA, EUA) solution and transferred to an IVD culture medium containing high-glucose Dulbecco’s Modified Eagle Medium (Gibco) supplemented with 5% foetal bovine serum (Sigma-Aldrich, St. Louis, MO, EUA), 1% penicillin/streptomycin (10,000 U/mL–10 mg/mL, Gibco), 0.5% amphotericin B (250 µg/mL, Sigma-Aldrich), 1% non-essential amino acids (Gibco) and 1.5% 5 mol/L NaCl/0.4 mol/L KCl solution to adjust the osmolarity of the medium to 400 mOsm. The IVDs were cultured in six-well plates with a membrane filter insert on top of each disc (Figure 1A) and under a static load of 0.46 MPa to avoid swelling [35], and at 37 °C in a reduced oxygen atmosphere (6% O_2_, 8.5% CO_2_) with saturated humidity. The medium exchange was performed every second day. At day 6, the IVDs were treated with (i) 17β-oestradiol (oestrogen, E2), (ii) LMHFV or (iii) both in combination (E2 + LMHFV). E2 was added to the medium at a concentration of 100 nM [27] and at every medium exchange. The groups treated with LMHFV were placed on a custom-made vibration platform (Figure 1B) and exposed to a frequency of 45 Hz and peak-to-peak acceleration of 0.3 g for 20 min per day, 5 days per week [28]. Samples were collected after 2 or 14 days of treatment (days 8 and 21 of organ culture, respectively) (Figure 1C). The timeline of the experiments is depicted in Figure 1D.

### 4.2. Gene Expression Analysis

Gene expression was assessed by qRT-PCR as previously described [52]. On day 8 of the organ cultures, tissue samples from the AF and NP regions were cut into small fragments, shock-frozen in liquid nitrogen and kept at −80 °C overnight. Subsequently, 1 mL RNAlater ICE (Invitrogen, Waltham, MA, EUA) was added to the tissues, which were stored at −20 °C. For the RNA isolation, the samples were thawed and homogenised in 1 mL QIAzol lysis reagent (Qiagen, Düsseldorf, Germany). For the two-phase extraction, 200 µL chloroform (Sigma-Aldrich) were added. Following incubation for 1 min, the samples were centrifuged at 20,000× *g* for 30 min at 4 °C. For the RNA isolation, the Arcturus PicoPure RNA isolation kit (Thermo Fisher Scientific) was used following the manufacturer’s instructions. The RNA concentration was determined using a Nano Quant plate and Tecan reader. The samples were kept on ice between steps. RNA was transcribed into cDNA using the Omniscript RT kit (Qiagen). After the cDNA was diluted 1:3 in nuclease-free water, qRT-PCR was performed using the custom-designed primers (Biomers, Ulm, Germany) in Table 1 and the Platinum SYBR Green qPCR SuperMix-UDG kit (Invitrogen), or TaqMan Gene Expression Assays and the Fast Advanced Master Mix (Applied Biosystems, Waltham, MA, EUA). Melt curve analysis to ensure assay specificity was performed and C_T_ values from samples with adequate PCR products were analysed following the 2^−ΔΔC_T_^ method [53,54]. For each target gene, the mean C_T_ value of each sample was normalised to the housekeeping gene glyceraldehyde 3-phosphate dehydrogenase (*GAPDH*) and to the control group, being ΔΔC_T_ = ΔC_T(sample of interest)_ − ΔC_T(control sample)_ and ΔC_T_ = C_T(gene of interest)_ − C_T(*GAPDH*)_.

### 4.3. Cell Viability

Prior to the TUNEL staining, IVD samples were flash-frozen in liquid nitrogen and stored at −80 °C. Cryosections of 10 μm were obtained and stained with 1 ng/mL Hoechst staining solution (Polysciences, Warrington, PA, USA) for 1 min and using the CF488A TUNEL apoptosis detection kit (Biotium, Fremont, CA, USA) according to the manufacturer’s instructions, as previously described [55]. Representative images were obtained from randomly selected regions of interest by fluorescence microscopy (Leica DMI6000B, Leica Microsystems, Wetzlar, Germany). Apoptotic cells were stained green, while cell nuclei were stained blue (Figure 3A). The percentage of cell viability was determined as: (number of blue-stained cells—number of green-stained cells)/number of blue-stained cells × 100.

### 4.4. PGE_2_ Quantification in Culture Supernatants

The concentration of PGE_2_ was measured by using an enzyme-linked immunosorbent assay kit (Arbor Assays, Ann Arbor, MI, USA) in IVD culture supernatants at days 8 and 21 following the manufacturers’ instructions.

### 4.5. DNA, sGAG and Collagen Quantification in AF and NP Tissues

At day 14, NP and AF samples were cut into small fragments, weighed and incubated overnight at 56 °C with 500 µL 0.5 mg/mL proteinase K solution for tissue digestion. The digests were stored at −20 °C for further analysis. DNA content in the AF and NP digests was determined using the Quant-iT PicoGreen dsDNA Assay kit. For sGAG quantification, the dimethylmethylene blue assay was applied as previously described [52,56]. Chondroitin sulphate (Sigma-Aldrich) was used to generate a standard curve. The absorbance was determined at 525 nm. A predictive model, generated by linear regression-fitting to the values of the standard curve, was utilised to determine the sGAG concentration. For hydroxyproline precipitation, 100 µL tissue digest were incubated with an equal volume of 37% hydrochloric acid (Sigma-Aldrich) for 24 h at 110 °C [57]. Following centrifugation at 10,000× *g* for 3 min, 2 µL of the supernatant were transferred to a 96-well plate and the solvent evaporated at 60 °C. The amount of hydroxyproline was determined following the kit’s instructions (Sigma-Aldrich). Following 90 min of incubation at 60 °C, the absorbance was determined at 560 nm. Considering that the hydroxyproline corresponds to approximately 10% of the mass of collagen [58,59], the collagen amount was extrapolated from the hydroxyproline measurements.

### 4.6. Histology

IVD tissues were fixed in 4% saline-buffered formalin for 48 h. Paraffin sections of 6 μm were cut in the sagittal plane. Safranin-O/fast green staining was performed for an overall assessment of the proteoglycan content of the tissue (stained pink/red). Briefly, sections were incubated in Weigert’s iron haematoxylin (Waldeck, Münster, Germany) for 5 min to stain the cell nuclei and, subsequently, immersed in 0.01% fast green (Waldeck) solution for 5 min to stain the collagen. Following rapid rinsing in 1% acetic acid (VWR, Radnor, PA, USA) solution, slides were immersed for 5 min in 0.1% safranin-O (Sigma-Aldrich) solution to detect proteoglycan deposition. The picrosirius red kit (Abcam, Cambridge, UK) was used to stain collagen. Sections were stained with picrosirius red solution for 1 h at room temperature. Birefringent collagen fibres were imaged with polarised light (Axiophot 451887, Zeiss, Jena, Germany). The colour hue corresponds to relative fibre thickness from thin green fibres to increasingly thick yellow, orange and red fibres. All images were captured with the same parameters. Area or red (1–9 nm; 230–255 nm), orange (10–38 nm), yellow (39–51 nm) and green (52–128 nm) fibres were quantified using ImageJ software following Pereira et al. (2016) [60].

### 4.7. Immunohistochemistry

Immunostaining was performed as previously described [52,55]. Paraffin sections were first incubated for 1 h at 37 °C and 30 min at 60 °C. The slides were deparaffinised in xylene (2× 100%) and descending ethanol solutions (2× 100%, 90%, 80%, 70%, 50%) and washed in distilled water. For unmasking, the sections were incubated in 10 mM citrate buffer at 95 °C for 20 min. Following cooling for 20 min, blocking was performed with 3% H_2_O_2_ in tris buffered saline (TBS) buffer for 20 min at room temperature and 5% goat serum in TBS with 0.1% Tween-20 (TBST) for 1 h at room temperature. Subsequently, incubation with primary antibody rabbit anti-IL-6 (5 µg/mL in TBST with 1% goat serum, Bioss, Woburn, MA, USA) or rabbit anti-MMP-3 (5 µg/mL in TBST with 1% goat serum, Abcam) was performed overnight at 4 °C. Following incubation with secondary antibody biotinylated goat anti-rabbit IgG (5 µg/mL dilution in TBST with 1% goat serum, Invitrogen) for 30 min, the VECTASTAIN Elite ABC HRP and the Vector NovaRED Substrate kits (both from Vector Laboratories, Newark, CA, USA) were used following the kits’ instructions. Cell nuclei were stained with haematoxylin. For dehydration, the sections were incubated in ascending ethanol solutions (90%, 100%) and xylene (2× 100%), and lastly covered with *Vitro*-*Clud* (R. Langenbrinck Labor- und Medizintechnik, Emmendingen, Germany). The stained sections were analysed following Saggese et al. (2019) [55]. Representative images were obtained under bright field microscopy. Artifacts were excluded by manually defining a region of interest using ImageJ software. Each image was deconvoluted into three different channels using a colour deconvolution plug-in for haematoxylin and 3,3′-diaminobenzidine (H DAB). The mean pixel intensity was quantified in the DAB channel. The data were normalised to control samples stained at the same time as samples from the additional groups [52,55].

### 4.8. Statistical Analysis

GraphPad Prism 8 (GraphPad Software, Inc, La Jolla, CA, USA) was used for the statistical analysis. Normal distribution was assessed with the Shapiro-Wilk test. For comparison of the normalised data to the control, the Wilcoxon signed rank test was performed. For parametric data, the comparison between groups was performed using Brown-Forsythe and Welch one-way analysis of variance, followed by Dunnett’s multiple comparison test. For nonparametric data, the comparison between groups was performed using Kruskal-Wallis test with Dunn’s multiple comparison test. Groups treated with E2 or LMHFV were compared to E2 + LMHFV. Differences were considered significant for *p* < 0.05.

## 5. Conclusions

While E2 and LMHFV are known to influence IVD homeostasis, this study provides further insights into their single and combined effect on IVD organ cultures. In this model, little effect of E2 on the IVD metabolism was observed, except for an increased production of IL-6 and MMP-3. Interestingly, LMHFV applied alone contributed to a decrease in cellularity and an increase in IL-6 production, but also in collagen, compared to the control. The combination of E2 + LMHFV seemed to counteract the effect of LMHFV alone by presenting a protective effect against cell loss and decreased IL-6 production compared to the LMHFV group. This ex vivo data suggests possible benefits of oestrogen therapy in combination with whole-body LMHFV on IVD homeostasis. However, further research is necessary to better understand the mechanism of action of E2 and LMHFV and their effect on IVDD.

## Figures and Tables

**Figure 1 ijms-24-06143-f001:**
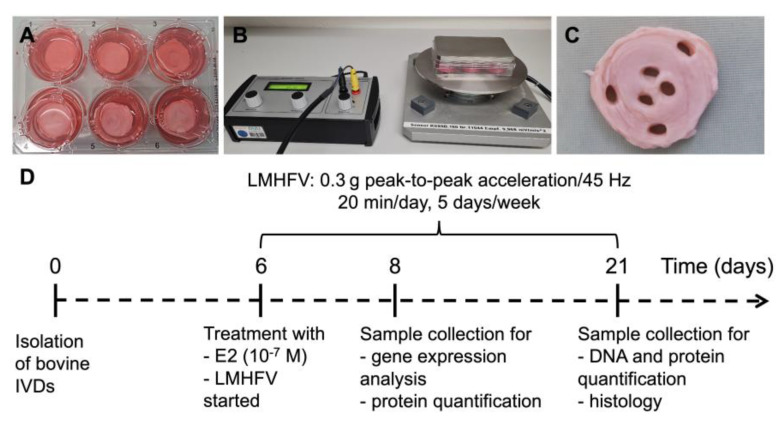
Bovine tail intervertebral disc organ culture experiments from day 0 until day 21. Treatment of the discs with 17β-oestradiol (E2, 100 nM), low-magnitude high-frequency vibration (LMHFV, 0.3 g peak-to-peak acceleration/45 Hz 20 min/day, 5 days/week) or both in combination (E2 + LMHFV) was initiated at day 6. (**A**) Isolated discs. (**B**) Custom-made vibration platform with discs under vibration. (**C**) Disc after removal of annulus fibrosus and nucleus pulposus tissue samples for further evaluations. (**D**) Timeline of the experiments.

**Figure 2 ijms-24-06143-f002:**
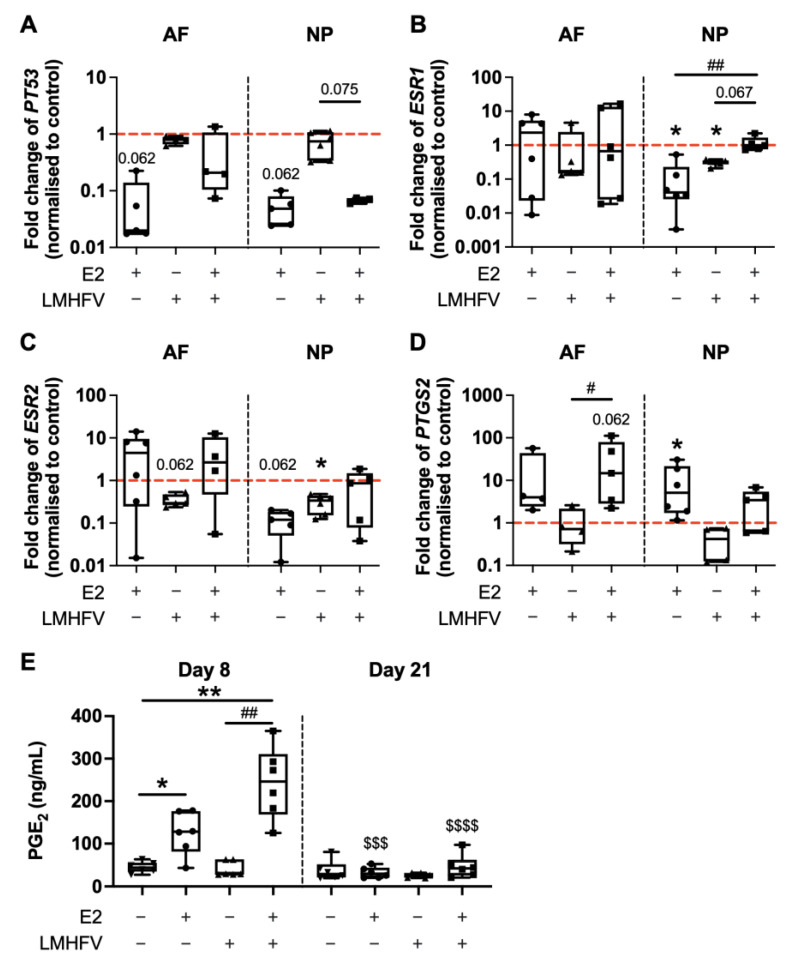
Relative gene expression of cell metabolism and apoptosis marker (**A**) *PT53*, oestrogen receptors (**B**) *ESR1* and (**C**) *ESR2*, and pro-inflammatory gene (**D**) *PTGS2* in cells of the AF and NP. RNA was isolated at day 8 of organ culture. IVD groups were either cultured under basal conditions (control) or treated with oestrogen (E2), low-magnitude high-frequency vibration (LMHFV) or the combination of E2 + LMHFV. Levels of mRNA were normalised to *GAPDH* and to the control (*red dashed line* = 1). Results are presented as 2^−ΔΔCt^. (**E**) PGE2 quantification (ng/mL) in culture supernatants at days 8 and 21. Box plots (*n* = 4–6 IVDs/group). * *p* < 0.05, ** *p* < 0.01 (*, comparison to control); ^#^ *p* < 0.05, ^##^ *p* < 0.01 (^#^, comparison to E2 + LMHFV group); ^$$$^ *p* < 0.001, ^$$$$^ *p* < 0.0001 (*, comparison to the same group at day 8).

**Figure 3 ijms-24-06143-f003:**
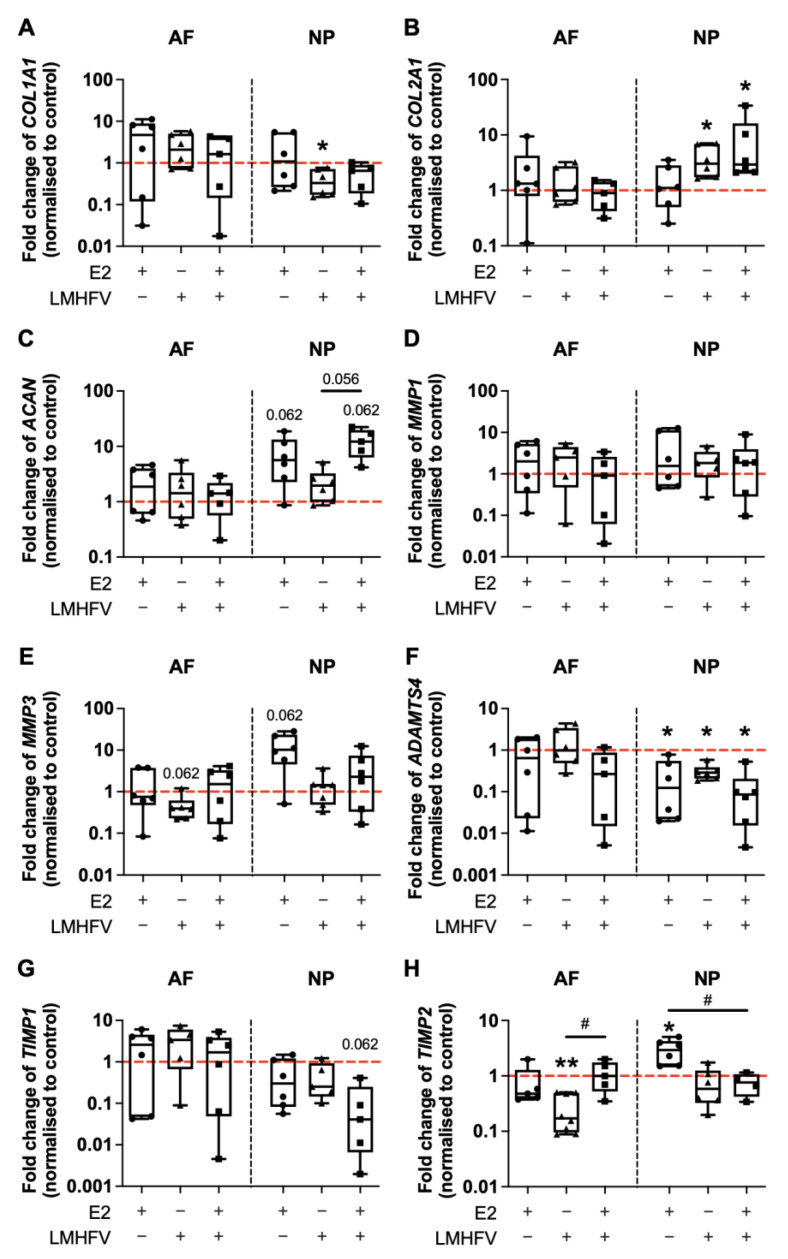
Relative gene expression of IVD matrix components (**A**) *COL1A1*, (**B**) *COL2A1* and (**C**) *ACAN*, matrix-degrading enzymes (**D**) *MMP1*, (**E**) *MMP3* and (**F**) *ADAMTS4* and inhibitors of matrix degradation (**G**) *TIMP1* and (**H**) *TIMP2* in cells of the AF and NP. RNA was isolated at day 8 of organ culture. IVD groups were either cultured under basal conditions (control) or treated with oestrogen (E2), low-magnitude high-frequency vibration (LMHFV) or the combination of E2 + LMHFV. Levels of mRNA were normalised to *GAPDH* and to the control (*red dashed line* = 1). Results are presented as 2^−ΔΔCt^ in box plots (*n* = 4–6 IVDs/group). * *p* < 0.05, ** *p* < 0.01 (*, comparison to the control); ^#^ *p* < 0.05 (^#^, comparison to E2 + LMHFV group).

**Figure 4 ijms-24-06143-f004:**
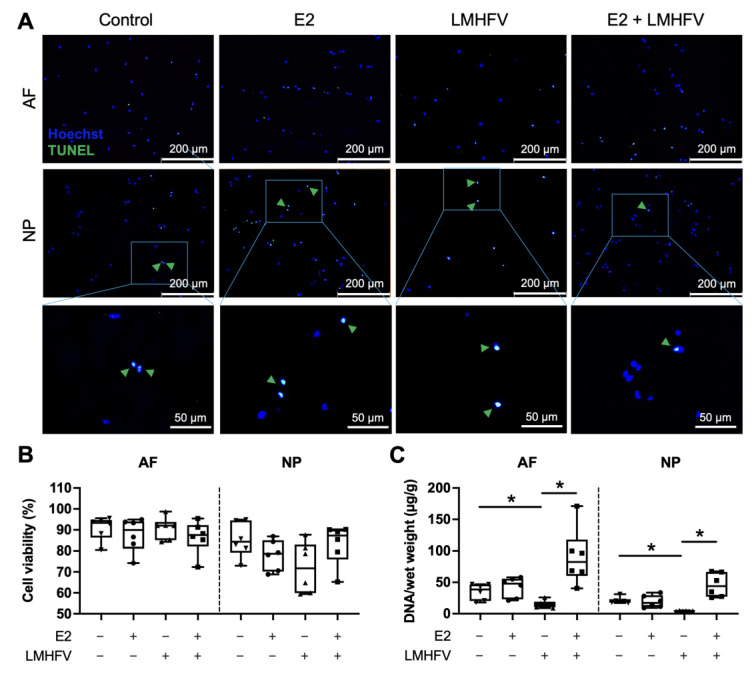
Cell viability in control, E2, LMHFV and E2 + LMHFV IVD tissues after 21 days of culture. (**A**) Representative fluorescence microscopy images of TUNEL staining in the AF and NP. Apoptotic cells are stained green (TUNEL, *green arrows*) and Hoechst counterstains cell nuclei blue (scale bars, 200 μm and 50 μm). (**B**) Quantification of the percentage of viable cells. (**C**) Amount of DNA normalised to wet weight (µg/g) in AF and NP tissues. Results are presented in box plots (*n* = 6 IVDs/group). * *p* < 0.05.

**Figure 5 ijms-24-06143-f005:**
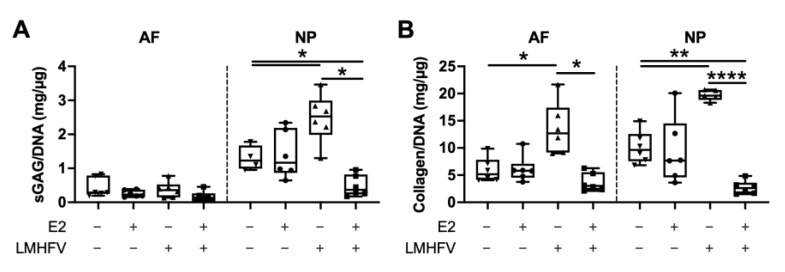
Protein quantification in AF and NP tissue digests of control, E2-, LMHFV- and E2 + LMHFV-treated IVD samples after 21 days of culture. (**A**) sGAG and (**B**) collagen normalised to DNA (mg/μg). Results are presented in box plots (*n* = 4–6 IVDs/group). * *p* < 0.05, ** *p* < 0.01, **** *p* < 0.0001.

**Figure 6 ijms-24-06143-f006:**
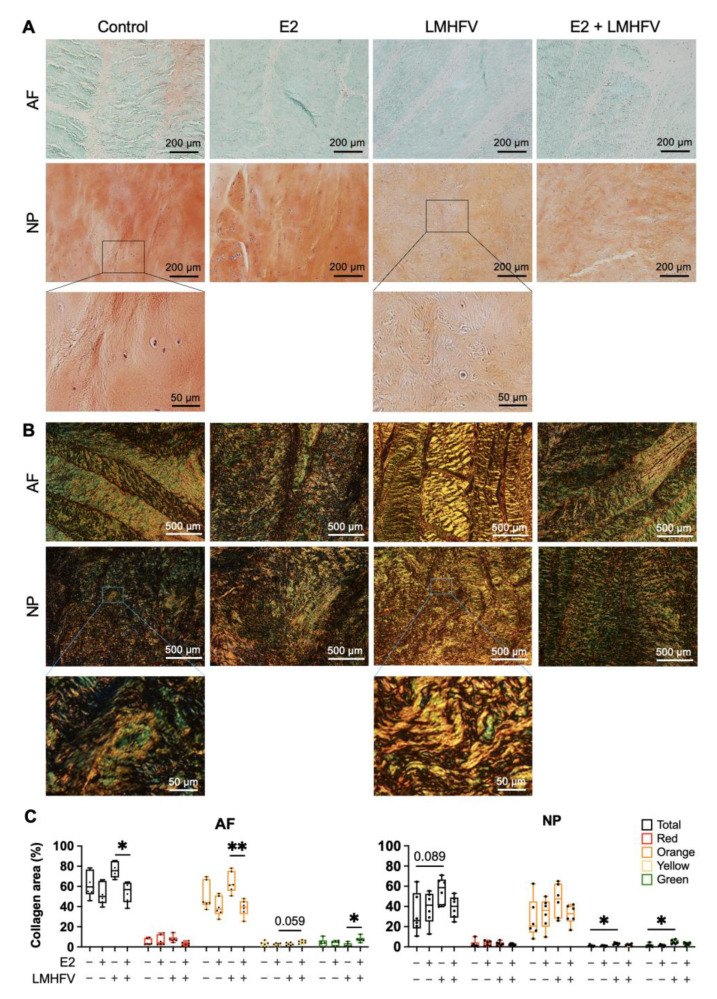
Histological analysis of control, E2, LMHFV and E2 + LMHFV IVD tissues after 21 days of culture. Sagittal IVD sections were stained with (**A**) safranin O/fast green for the detection of proteoglycans (proteoglycans are stained orange and collagens are counterstained green; scale bar, 200 μm) or (**B**) picrosirius red for the detection of collagens. The colour range visualised under polarised light microscopy corresponds to relative fibre thickness from thin and less mature green fibres to increasingly thicker and more mature yellow, orange and red fibres (scale bar, 500 μm). AF and NP regions of one representative donor are depicted. (**C**) Quantification of the relative percentage of collagen fibres in the AF and NP regions of the picrosirius red-stained sections. Results are presented in box plots (*n* = 6 IVDs/group). * *p* < 0.05, ** *p* < 0.01.

**Figure 7 ijms-24-06143-f007:**
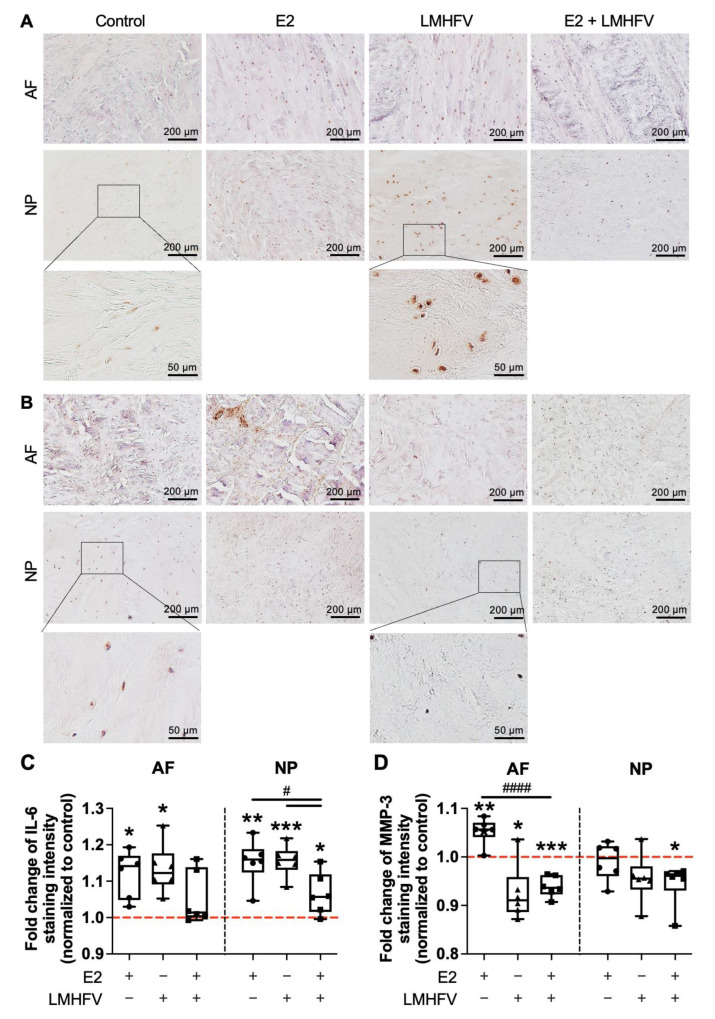
Representative images of AF and NP sagittal sections stained for (**A**) IL-6 and (**B**) MMP-3 in the control, E2, LMHFV and E2 + LMHFV groups after 21 days of IVD culture (scale bar, 200 μm). Quantification of (**C**) IL-6 and (**D**) MMP-3 fluorescence intensity normalised to control (red *dashed line* = 1). Results are presented in box plots (*n* = 6 IVDs/group). * *p* < 0.05, ** *p* < 0.01, *** *p* < 0.001 (*, comparison to the control); ^#^ *p* < 0.05, ^####^ *p* < 0.0001 (^#^, comparison to the E2 + LMHFV group).

**Figure 8 ijms-24-06143-f008:**
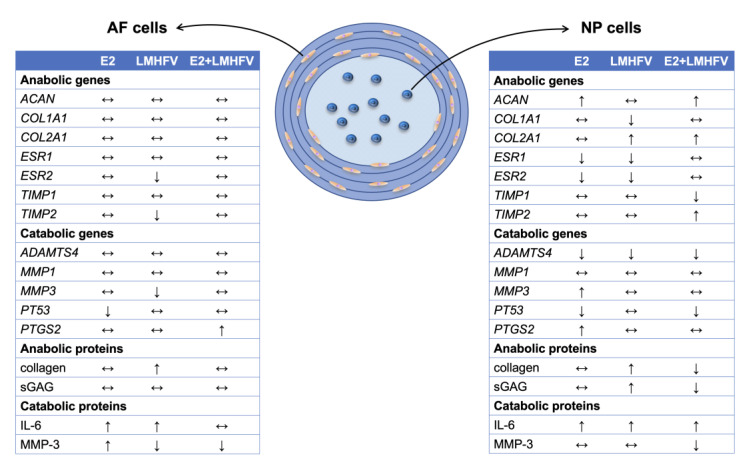
Regulation of anabolic and catabolic genes in AF and NP cells at day 8 of organ culture, and of proteins at day 21 by E2, LMHFV and E2 + LMHFV treatments compared to the unstimulated control group. ↑, upregulation/increase; ↓, downregulation/decrease; ↔, no alteration.

**Table 1 ijms-24-06143-t001:** Bovine oligonucleotide primers used for qRT-PCR. Primers with a shown sequence were custom designed; primers with an assay ID number were purchased from Applied Biosystems. fw: forward; rev: reverse.

Gene	Sequence (Forward and Reverse Primer)	Product Size (bp)
*ACAN*	fw: 5′-ACA GCG CCT ACC AAG ACA AG-3′ rev: 5′-ACG ATG CCT TTT ACC ACG AC-3′	155
*COL1A1*	fw: 5′-TGA GAG AGG GGT TGT TGG AC-3′ rev: 5′-AGG TTC ACC CTT CAC ACC TG-3′	142
*COL2A1*	5′-CCT GTA GGA CCT TTG GGT CA-3′ 5′-ATA GCG CCG TTG TGT AGG AC-3′	145
*ESR1*	fw: 5′-GCC TCA AAT CCA TCA TCT TGC T-3′ rev: 5′-CGG TGG ATG TGG TCC TTC TC-3′	100
*ESR2*	fw: 5′-CTC CTG GAC ACC TCT CTC CTT TAG-3′ rev: 5′-GGT TTC ACG CCA AGG ACT CTT-3′	85
*GAPDH*	fw: 5′-ACC CAG AAG ACT GTG GAT GG-3′ rev: 5′-CAA CAG ACA CGT TGG GAG TG-3′	178
*MMP1*	fw: 5′-ATG CTG TTT TCC AGA AAG GTG G-3′ rev: 5′-TCA GGA AAC ACC TTC CAC AGA C-3′	193
*MMP3*	fw: 5′-CTG CGG ATA CTT CCA CAG GT-3′ rev: 5′-ATG GAT GAG CAG GGA AAC AC-3′	198
*PT* *53*	fw: 5′-ATT TAC GCG CGG AGT ATT TG-3′ rev: 5′-CCA GTG TGA TGA TGG TGA GG-3′	174
*PTGS2*	Bt03214492_m1	
*TIMP1*	fw: 5′-GCT GGA CAT TGG AGG AAA GA-3′ rev: 5′-CGT CCG GAG AGG AGA TGT AG-3′	209
*TIMP2*	fw: 5′-TGA GAG AGG GGT TGT TGG AC-3′ rev: 5′-AGG TTC ACC CTT CAC ACC TG-3′	142

## Data Availability

Not applicable.

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
