# Peer review of "Oestrogen and Vibration Improve Intervertebral Disc Cell Viability and Decrease Catabolism in Bovine Organ Cultures"

_ijms, 2023, doi:10.3390/ijms24076143_

Round 1

Reviewer 1 Report

The manuscript entitled“Oestrogen and vibration improve intervertebral disc cell viability and decrease catabolism in bovine organ cultures ” focuses on whether oestrogen has a protective effect on intervertebral disc degeneration, and concludes that E2-replacement therapy is possibly beneficial for the IVDs of postmenopausal women undergoing LMHFV exercises. This study may provide new insights into the treatment of IDD in postmenopausal women. 

Some comments:

10−7 M is not a common used format to indicate concentrations, suggest to use uM, nM etc.

In Figure 2, no E2-LMHFV- group has been set for ESR1 and ESR2,PTGS2 and P53 expression. It is suggested that the experimental design should be improved with E2-LMHFV- group as control, so as to firstly confirm the effect of E2 or LMHFV alone on IVD cells, as well as confirm the modulation of oestrogen receptors of E2 on IVD cells.

It is unclear why the IHC staining photos in Figure 6A and Figure 6B both have control groups, but Figure 6C, the quantitative analysis of Figure 6A and 6B, has no control group. It is suggested to include the control group into quantificative analysis.

The sentence 'The combination of E2 + LMHFV induced a protective effect against cell loss and decreased MMP-3 and IL-6 production' is not accurate. Without specifying the counterpart group, this sentence may be easily interpreted by the readers that E2 + LMHFV induced a protective effect against cell loss and decreased MMP-3 and IL-6 production compared to nil treatment. However, the true message is that LMHFV alone induced cell loss and increased MMP-3 and IL-6 production compared to the control, while E2 could counteract this effect. Therefore, it is suggested to add 'compared to LMHFV group' in the mentioned sentence.

Author Response

Point-by-point reply to Reviewer 1

The manuscript entitled “Oestrogen and vibration improve intervertebral disc cell viability and decrease catabolism in bovine organ cultures ” focuses on whether oestrogen has a protective effect on intervertebral disc degeneration, and concludes that E2-replacement therapy is possibly beneficial for the IVDs of postmenopausal women undergoing LMHFV exercises. This study may provide new insights into the treatment of IDD in postmenopausal women. 

Authors: We would like to thank the reviewer for the assessments. We hope to have adequately addressed all comments and recommendations. The revised version of the manuscript is submitted in attachment. Changes are highlighted on the manuscript in bold and red.

Some comments:

  1. 10−7 M is not a common used format to indicate concentrations, suggest to use uM, nM etc.

Authors: We have changed it to 100 nM.

  1. In Figure 2, no E2-LMHFV- group has been set for ESR1 and ESR2,PTGS2 and P53 expression. It is suggested that the experimental design should be improved with E2-LMHFV- group as control, so as to firstly confirm the effect of E2 or LMHFV alone on IVD cells, as well as confirm the modulation of oestrogen receptors of E2 on IVD cells.

Authors: Figure 2 shows the results for E2, LMHFV and for E2+LMHFV normalized to the control group (represented by the red dashed line = 1), corresponding the first column to E2, the second to LMHFV and the third to E2+LMHFV. For the statistical analysis, all treatment groups were compared to the control. Additionally, E2 and LMHFV alone were also compared to E2+LMHFV. The data is presented following the 2(-ΔΔCt) method (Bachmeier et al., Exp Cell Res, 2005; Livak and Schmittgen, Methods, 2001), being ΔΔCT = ΔCT(sample of interest) - ΔCT(control sample) and ΔCT = CT(gene of interest) - CT(GAPDH). We added this information to the Material and Methods section, lines 434-435. The title of the yy axis was modified in all graphs from Figure 2 (and Figure 3) to include “(normalised to control)”.

  1. It is unclear why the IHC staining photos in Figure 6A and Figure 6B both have control groups, but Figure 6C, the quantitative analysis of Figure 6A and 6B, has no control group. It is suggested to include the control group into quantificative analysis.

Authors: The authors apologize if the legends of the graphs in Figures 6C and 6D were not clear. Determination of staining intensity is a semi-quantitative method; therefore, the staining intensity for each sample from E2, LMHFV and E2+LMHFV was normalized to the average value of the control samples (represented by the red dashed line = 1) stained at the same time. The method is briefly described in lines 508-512 and was previously published in Neidlinger-Wilke et al., Eur Cell Mater, 2021 and Saggese et al., Eur Spine J, 2019. The title of the yy axis was modified in all graphs from Figure 6 to include “(normalised to control)”.

  1. The sentence 'The combination of E2 + LMHFV induced a protective effect against cell loss and decreased MMP-3 and IL-6 production' is not accurate. Without specifying the counterpart group, this sentence may be easily interpreted by the readers that E2 + LMHFV induced a protective effect against cell loss and decreased MMP-3 and IL-6 production compared to nil treatment. However, the true message is that LMHFV alone induced cell loss and increased MMP-3 and IL-6 production compared to the control, while E2 could counteract this effect. Therefore, it is suggested to add 'compared to LMHFV group' in the mentioned sentence.

Authors: The authors acknowledge the Reviewer’s comment have rephrased the text in lines 21-24, as suggested.

Reviewer 2 Report

The authors have investigated the method of Low-magnitude high-frequency vibration in a IVD bovine model and found the genetic factors that are being altered. The research is interesting and has been designed to satisfy questions raised in back pain therapy. However, the authors need to modify and explain further a few topics before the journal paper is found suitable for publication.

1. The figures need to be represented in a way that is much more comprehensible especially figure 2 and figure 3. 

2. A magnified image of figure 4 will help in providing impact to the authors work.

3. The discussion needs to have more references to build on the work done. 

Author Response

Point-by-point reply to Reviewer 2

The authors have investigated the method of Low-magnitude high-frequency vibration in a IVD bovine model and found the genetic factors that are being altered. The research is interesting and has been designed to satisfy questions raised in back pain therapy. However, the authors need to modify and explain further a few topics before the journal paper is found suitable for publication.

Authors: We would like to thank the reviewer for the assessments. We hope to have adequately addressed all recommendations. The revised version of the manuscript is submitted in attachment. Changes are highlighted on the manuscript in bold and red.

  1. The figures need to be represented in a way that is much more comprehensible especially figure 2 and figure 3. 

Authors: We have modified the graphs in all figures (except new Figure 6) combining in one graph the data from AF and NP. We hope this modification made them more comprehensible.

  1. A magnified image of figure 4 will help in providing impact to the authors work.

Authors: We have included higher magnification examples in Figure 4.

  1. The discussion needs to have more references to build on the work done. 

Authors: We did an additional extensive literature research to include all relevant work. We tried to cite all the available references regarding the influence of E2 and vibration on IVD cells; however, there is not much literature available particularly for low-magnitude high-frequency vibration (LMHFV), showing that this topic is under-investigated. To the best of our knowledge, this was the first work investigating the effect of the E2 + LMHFV on the IVD.

Reviewer 3 Report

This paper described the effect of 17β-oestradiol (E2) and Low-magnitude high-frequency vibration (LMHFV) on gene expression, cellularity and matrix changes in intervertebral disc (IVD)  ex vivo. The authors found that a combination of E2 of LMHFV induced a protective effect against cell loss and production of MMP-3 and IL-6. These findings imply a possible therapeutical method for IVD degeneration. However, there are several issues in the experimental strategies (discussed in more detail below). 

1. The author claims that there may be differences between male and female IVDD patients (Line 40). However, the authors used male bovine IVD to perform the experiment. The reviewers wonder the rationale of using male IVD instead of female ones. Are the structure and cellular and molecular composition, especially estrogen receptors, the same?

2. The authors used healthy bovine disc to evaluate the effects of E2 and LMHFV. The finding may be different in post-menopausal osteoporotic samples.

3. The study showed LMHFV alone led to diminished cellularity and an increase in IL-6 while combination of E2 + LMHFV maintained cellularity and decreased MMP-3 and IL-6 production. These data suggest E2 is indispensable for the best results, which is contradict to the authors’ claim that “This implicates possible benefits of E2-replacement therapy for the IVDs of postmenopausal women undergoing LMHFV exercises. (Line 24-25)”

4. According to Line 70-71, the authors clarified that LMHFV played a paradoxical role in IVD metabolism. It seemed to be more relevant for combined E2 therapy which explored the most suitable parameters for IVD metabolism.

5. According to Line 64-66,whole-body LMHFV (0.3 g peak-to-peak acceleration/45 Hz, 20 min/day, 5 days/week) was also shown to induce meniscal tears and articular cartilage damage. In Fig4, the results suggested that LMHFV reduced AF, NP normalized DNA content while E2 reverses LMHFV damage. These data reflected the important intervening factor in reducing cell loss not by LMHFV but E2.

6. In Fig4, the control group as well as E2 group had about 40% orange collagen. The percentage of orange collagen was increased in the LMHFV group while those one was close to control group in E2 and LMHFV combination. It seemed to break the ratio of mature collagen.

7. In Fig2, Fig 3, Fig 6, the control group data always were lack.

8. The study was largely observatory, which lacks mechanism of action.

Author Response

Point-by-point reply to Reviewer 3

This paper described the effect of 17β-oestradiol (E2) and Low-magnitude high-frequency vibration (LMHFV) on gene expression, cellularity and matrix changes in intervertebral disc (IVD) ex vivo. The authors found that a combination of E2 of LMHFV induced a protective effect against cell loss and production of MMP-3 and IL-6. These findings imply a possible therapeutical method for IVD degeneration. However, there are several issues in the experimental strategies (discussed in more detail below). 

Authors: We would like to thank the reviewer for the assessments. We hope to have adequately addressed all comments and recommendations. The revised version of the manuscript is submitted in attachment. Changes are highlighted on the manuscript in bold and red.

  1. The author claims that there may be differences between male and female IVDD patients (Line 40). However, the authors used male bovine IVD to perform the experiment. The reviewers wonder the rationale of using male IVD instead of female ones. Are the structure and cellular and molecular composition, especially estrogen receptors, the same?

Authors: The authors used male bovine discs because of more availability in comparison to female ones, since male cows are killed at younger age for food, whereas females are kept longer for milk production. However, we do not expect significant differences between sex regarding ex vivo findings. The experiments were conducted in organ cultures and, therefore, without physiological influence of hormones such as oestrogen. Of course, the situation might be different in an in vivo model (or the clinical situation), where the systemic influence of oestrogen and other sex hormones is very prominent. Regarding the expression of ERs, there might be some slight differences also in in vitro systems. ERα and ERβ have both been detected in human AF (Gruber et al., BMC Musculoskelet Disord, 2002) and NP cells and their expression was shown to be significantly decreased in the NP with the aggravation of IVD degeneration both in samples from male and female patients (Song et al., Joint Bone Spine, 2014; Song et al., Oncotarget, 2017). It has also been shown that the expression of ERα and ERβ proteins in the NP tissue of degenerated IVDs of males is significantly higher than that of females (Song et al., Joint Bone Spine, 2014). In conclusion, we think that the effect of sex might be neglectable in our in vitro system, which might be different in an in vivo situation. However, clearly further work is needed to proof that. We included this in the discussion, lines 273-284.

  1. The authors used healthy bovine disc to evaluate the effects of E2 and LMHFV. The finding may be different in post-menopausal osteoporotic samples. 

Authors: In particular, elderly females experience a faster rate of lumbar disc degeneration with greater severity than age-matched men, despite the frequency of lumbar disc degeneration being higher in young and middle-aged men than in women (Takatalo et al., Spine, 2009; Wang and Griffith, Radiology, 2010). However, before investigating the effect of E2 and LMHFV in a model of disc degeneration, our focus was to better understand their effects on the matrix metabolism of heathy/mildly degenerated IVDs. This because, previous finding in bone research showed a positive effect of LMHFV on fracture healing repair in ovariectomized mice (which were not capable of produce oestrogen), whereas in non-ovariectomized heathy mice, LMHFV negatively affected bone repair (Haffner-Luntzer et al., Bone, 2018). We have included a sentence to the discussion, lines 364-368.

  1. The study showed LMHFV alone led to diminished cellularity and an increase in IL-6 while combination of E2 + LMHFV maintained cellularity and decreased MMP-3 and IL-6 production. These data suggest E2 is indispensable for the best results, which is contradict to the authors’ claim that “This implicates possible benefits of E2-replacement therapy for the IVDs of postmenopausal women undergoing LMHFV exercises. (Line 24-25)”

Authors: The authors acknowledge the reviewer’s comment. “E2-replacement therapy” was rephrased to “oestrogen therapy” (line 25).

  1. According to Line 70-71, the authors clarified that LMHFV played a paradoxical role in IVD metabolism. It seemed to be more relevant for combined E2 therapy which explored the most suitable parameters for IVD metabolism.

Authors: Both the effects of LMHFV and E2 on intervertebral disc metabolism are under investigated in the literature. Our work showed that LMHFV effects may be dependent (among other factors) on the presence or absence of E2. Indeed, application of E2 + LMHFV seemed to have a positive effect on IVD metabolism. The possible use of E2 + LMHFV as combined therapy, for instance, for intervertebral disc degeneration will be the focus of future mechanistic investigations. We have rephrased it the discussion, lines 350-374.

  1. According to Line 64-66,whole-body LMHFV (0.3 g peak-to-peak acceleration/45 Hz, 20 min/day, 5 days/week) was also shown to induce meniscal tears and articular cartilage damage. In Fig4, the results suggested that LMHFV reduced AF, NP normalized DNA content while E2 reverses LMHFV damage. These data reflected the important intervening factor in reducing cell loss not by LMHFV but E2. 

Authors: The authors knowledge the reviewer’s comments and have reinforced this statement in the discussion section (lines 359-368). Vibration platforms that apply whole-body LMHFV are often used for physical exercise or to accelerate bone healing after fracture. However, the work form Haffner-Luntzer et al., Bone, 2018 showed only a positive effect of LMHFV on fracture healing repair in ovariectomized female mice (which were not capable of produce oestrogen), whereas in non-ovariectomized ones, LMHFV negatively impacted bone repair. Our data showed that also in the IVD, the effect of LMHFV seems to be E2-dependent and that E2, more than LMHFV, contributes to IVD reduced cell loss. This is in agreement with previous studies showing that E2 can promote AF and NP cell proliferation by reducing the level of apoptosis in vitro (Gruber et al., BMC Musculoskelet Disord, 2002; Yang et al., Apoptosis, 2014; Yang et al., Apoptosis, 2015).

  1. In Fig4, the control group as well as E2 group had about 40% orange collagen. The percentage of orange collagen was increased in the LMHFV group while those one was close to control group in E2 and LMHFV combination. It seemed to break the ratio of mature collagen.

Authors: Alterations in collagen glycosylation have been implicated in fibrillar collagen maturation (Terajima et al., J Biol Chem, 2014). Given the increase of orange “more-mature” collagen fibers in the AF of LMHFV-treated IVDs, we hypothesize that LMHFV has an impact on collagen glycosylation promoting maturation. Future investigations will explore this hypothesis. We have included this to the discussion, lines 339-343.

  1. In Fig2, Fig 3, Fig 6, the control group data always were lack.

Authors: The authors apologize if the legends of the graphs in Figures 2, 3 and 6 were not clear. The data was normalized to the control group (represented by the dashed line = 1). The title of the yy axis was modified in all graphs to include “(normalized to control)”. The gene expression data in Figures 2 and 3 is presented following the 2(-ΔΔCt) method (Bachmeier et al., Exp Cell Res, 2005; Livak and Schmittgen, Methods, 2001), being ΔΔCT = ΔCT(sample of interest) - ΔCT(control sample) and ΔCT = CT(gene of interest) - CT(GAPDH). We added this information to the Material and Methods section, lines 432-435. Additionally, for Figure 6, the staining intensity for each sample of interest (i.e., from E2, LMHFV and E2+LMHFV) was normalized to the average value of the control samples stained at the same time. The method is described in lines 508-512 and was previously published in Neidlinger-Wilke et al., Eur Cell Mater, 2021 and Saggese et al., 2019.

  1. The study was largely observatory, which lacks mechanism of action.

Authors: The purpose of the present work was to better understand whether E2 and/or LMHFV have an effect on the matrix metabolism of heathy/mildly degenerated IVDs ex vivo. Mechanistic investigations will be included in future experiments. However, the discussion section was rephrased to include literature in which the mechanism of action of E2 and LMHFV was investigated.

Reviewer 4 Report

The present work site describe an in-vitro experiment  using bovine intervertebral disc as a model or IVDD. They show a series of analysis on IVD cultivate under presence, co.bi ed or alone,  of E2 or LMHFV. Using a mainly  combination of QPCR qnd imunohistochemestry they show a potencial benefit of E2 and LMHFV to help treat IVDD. The evidence in the present format is interesting and sound to enhance future studies in the field.

The article shows evidence that Low-magnitude high-frequency vibration (LMHFV) associated or not with oestradiol can reduce the markers of apoptosis, and influence collagen and IL-6 formation, The effects of Oestradiol and LMHFV were already known to be beneficial but described results increase the evidence and helps to understand the mechanism behind those effects.

The conclusions consistent with the evidence and arguments presented but perhaps too bold for an in-vitro assay, although, it is expected to extrapolate basic and in-vitro assay results in order to justify the possible outcomes from the initial results.

Author Response

Point-by-point reply to Reviewer 4

The present work site describe an in-vitro experiment  using bovine intervertebral disc as a model or IVDD. They show a series of analysis on IVD cultivate under presence, co.bi ed or alone,  of E2 or LMHFV. Using a mainly  combination of QPCR qnd imunohistochemestry they show a potencial benefit of E2 and LMHFV to help treat IVDD. The evidence in the present format is interesting and sound to enhance future studies in the field.

The article shows evidence that Low-magnitude high-frequency vibration (LMHFV) associated or not with oestradiol can reduce the markers of apoptosis, and influence collagen and IL-6 formation, The effects of Oestradiol and LMHFV were already known to be beneficial but described results increase the evidence and helps to understand the mechanism behind those effects.

The conclusions consistent with the evidence and arguments presented but perhaps too bold for an in-vitro assay, although, it is expected to extrapolate basic and in-vitro assay results in order to justify the possible outcomes from the initial results.

Authors: We would like to thank the reviewer for the assessments. We have rephrased the conclusions in lines 375-385 and lines 523-533. The revised version of the manuscript is submitted in attachment. Changes are highlighted on the manuscript in bold and red.

Reviewer 5 Report

The study “Oestrogen and vibration improve intervertebral disc cell viability and decrease catabolism in bovine organ cultures” is interesting and potentially clinically relevant.

 1)The meaning of E2 and LMHFV is not introduced in the abstract. Please correct.

2)For all the results, please indicate the magnitude of the difference in actual numbers or fold change in addition to mentioning if it is statistically different. It is important to know if the differences are of biological relevance.

3)The graphs are very small and hard to follow. Please adjust.

 4)Figures 4-6 Please insert higher mag inserts where the staining can be appreciated.

5)Was GAG in the culture media evaluated? If culture was done without endplates, it is likely that GAG was lost as a result of the culture system and that newly synthesized GAG failed to get incorporated and was lost from the tissue.  How did GAG content compare to freshly isolated tissue? Please address.

6)What is the baseline expression of ESR1 and ESR2?

7)For the discussion, do not repeat the result section, remove the referral to figures and propose a meaning of the results.

8)Please provide a reference for “E2 modulation of these pro-inflammatory cytokines was shown to be mediated by substance P”.

9)It is written, “Less cells to produce ECM may explain the lower amount of sGAG in the NP, but not the increase in collagen production (Figure 5)”. Similar to the comment above, GAG can easily diffuse out of the tissue, whereas collagen cannot. Please rephrase the conclusion.

10)Justify why male IVDs were treated with estrogen if it is proposed to help women.

11)Static loading of 0.46 MPa was applied to avoid swelling. Please indicate the magnitude of IVD height loss.  This is a high load that has been shown to induce substantial cell death in bovine IVDs. Is the drastic cell death because of a too-high load? 

12)Cell death was evaluated by measuring apoptosis. How were cells that didn’t die through apoptosis accounted for and how was the cell density (or total DNA) compared to day 0? Please provide data.

Author Response

Point-by-point reply to Reviewer 5

The study “Oestrogen and vibration improve intervertebral disc cell viability and decrease catabolism in bovine organ cultures” is interesting and potentially clinically relevant.

Authors: We would like to thank the reviewer for the assessments. We hope to have adequately addressed all comments and recommendations. The revised version of the manuscript is submitted in attachment. Changes are highlighted on the manuscript in bold and red.

1) The meaning of E2 and LMHFV is not introduced in the abstract. Please correct.

Authors: We would like to thank the reviewers for the assessments. The meaning of LMHFV is introduced in lines 14-15. E2 is described in line 17.

2) For all the results, please indicate the magnitude of the difference in actual numbers or fold change in addition to mentioning if it is statistically different. It is important to know if the differences are of biological relevance.

Authors: The authors have included the fold change information to the results section.

3) The graphs are very small and hard to follow. Please adjust.

Authors: We have combined in one graph the data from AF and NP. We have also increased the letter size of the graphs and reorganized the images. We hope this modification made them more comprehensible.

 4) Figures 4-6 Please insert higher mag inserts where the staining can be appreciated.

 Authors: Higher magnification examples were included in Figures 4-6.

5) Was GAG in the culture media evaluated? If culture was done without endplates, it is likely that GAG was lost as a result of the culture system and that newly synthesized GAG failed to get incorporated and was lost from the tissue.  How did GAG content compare to freshly isolated tissue? Please address.

Authors: The authors acknowledge the reviewer’s comment. We have included in the Appendix A (Figure A4A) the quantification of sGAG in freshy isolated IVD tissues. About 16±5 and 55±10 mg of sGAG/g of IVD tissue wet weight were detect in AF and NP samples, respectively. This data is in agreement with the literature (Bezci et al., JOR Spine, 2019; Fiordalisi et al., Biomater Adv, 2022). Significantly lower sGAG was observed in the organ culture controls at day 28 than in fresh AF and NP samples (0.02±0.71- and 0.40±0.32-fold, respectively, p<0.01, Figure A4A). GAG loss in bovine IVD organ cultures without endplates with time in culture has been previously observed by us (Teixeira et al., Tissue Eng Part C Methods, 2016) and others (Korecki et al., Eur Spine J, 2007; Li et al., Spine, 2006). However, in the present work, when comparing the GAG loss to the culture supernatant at days 8 and 21, no significant differences were observed between the different groups (Appendix A, Figure A4B). As previously shown in Teixeira et al., Tissue Eng Part C Methods, 2016 (supplemental data), the strongest loss occurs within the first 6-10 days of organ culture. This was added to the discussion, lines 262-266.

6) What is the baseline expression of ESR1 and ESR2?

Authors: We included the data for the expression of ESR1 and ESR2 from freshly isolated AF and NP tissues to the Appendix A, Figure A1. The results are presented following the 2(-ΔCt) method (Bustin et al., 2009), being ΔCT = CT(gene of interest) - CT(GAPDH). The data was compared to the expression of ESR1 and ESR2 after 8 days of organ culture under control conditions. Results showed a significant upregulation of ESR1, but not ESR2, by AF and NP cells with time in culture (113.2 ± 1.1- and 8.5 ± 1.2-fold, respectively, p<0.01, Figure A1). A sentence was also included in the discussion, lines 257-261.

7) For the discussion, do not repeat the result section, remove the referral to figures and propose a meaning of the results.

 Authors: The authors removed the referral to figures and rephrased the discussion section.

8) Please provide a reference for “E2 modulation of these pro-inflammatory cytokines was shown to be mediated by substance P”.

Authors: The respective references were added to the sentence (lines 309-311).

9) It is written, “Less cells to produce ECM may explain the lower amount of sGAG in the NP, but not the increase in collagen production (Figure 5)”. Similar to the comment above, GAG can easily diffuse out of the tissue, whereas collagen cannot. Please rephrase the conclusion.

Authors: We thank the reviewer for the comment and have rephrased the sentence in lines 337-339. 

10) Justify why male IVDs were treated with estrogen if it is proposed to help women.

Authors: The authors used male bovine discs because of more availability in comparison to female ones, since male cows are killed at younger age for food, whereas females are kept longer for milk production. However, we do not expect significant differences between sex regarding ex vivo findings. The experiments were conducted in organ cultures and, therefore, without physiological influence of hormones such as oestrogen. Of course, the situation might be different in an in vivo model (or the clinical situation), where the systemic influence of oestrogen and other sex hormones is very prominent. Regarding the expression of ERs, there might be some slight differences also in in vitro systems. ERα and ERβ have both been detected in human AF (Gruber et al., BMC Musculoskelet Disord, 2002) and NP cells and their expression was shown to be significantly decreased in the NP with the aggravation of IVD degeneration both in samples from male and female patients (Song et al., Joint Bone Spine, 2014; Song et al., Oncotarget, 2017). It has also been shown that the expression of ERα and ERβ proteins in the NP tissue of degenerated IVDs of males is significantly higher than that of females (Song et al., Joint Bone Spine, 2014). In conclusion, we think that the effect of sex might be neglectable in our in vitro system, which might be different in an in vivo situation. However, clearly further work is needed to proof that. We included this in the discussion, lines 273-284.

11) Static loading of 0.46 MPa was applied to avoid swelling. Please indicate the magnitude of IVD height loss.  This is a high load that has been shown to induce substantial cell death in bovine IVDs. Is the drastic cell death because of a too-high load? 

Authors: IVD tissues with 5 mm height were isolated from the caudally and cranially adjacent cartilaginous endplates using a custom-built cutting-tool, containing two parallel microtome blades 5 mm apart. This information was added to lines 395-397. At days 8 and 21 of the experiments, the height of the IVDs was about 4.2±0.3 mm, indicating a loss of about 15% of the initial IVD height. For instance, the work from Li and colleagues (Front Bioeng Biotechnol, 2020) showed that on day 3 of bovine disc organ culture (with endplates), directly after application of dynamic loading (0.32–0.5 MPa; 5 Hz; 2 h/day), the temporary IVD height loss was over 20%, with a decrease in cell viability of about 50% in the inner AF. Although in vivo measurements have shown that the inner lumbar disc pressure during relaxed standing is about 0.5 MPa (Wilke et al., Spine, 1999), indeed, in the present work, a significant reduction of DNA content/cell viability was observed in AF and NP samples from the control group at day 21 when compared to fresh ones (0.3±0.4- and 0.5±0.3-fold, respectively, p<0.05, Figure A3), in agreement with the work form Li and colleagues. However, we hypothesize that the observed cell death is associated not with the loading magnitude (which is within the normal physiological range), but with the fact that a constant static loading was applied instead of dynamic loading, which may have limited fluid-flow and nutrition of the inner AF and NP. This was addressed in the discussion section, lines 262-266.

12) Cell death was evaluated by measuring apoptosis. How were cells that didn’t die through apoptosis accounted for and how was the cell density (or total DNA) compared to day 0? Please provide data.

Authors: Previous studies showed, for instance, that E2 mostly modulates AF and NP cell viability by reducing apoptosis in vitro (Gruber et al., BMC Musculoskelet Disord, 2002; Yang et al., Apoptosis, 2014; Yang et al., Apoptosis, 2015), therefore, we did not perform further assays than TUNEL and DNA quantification. However, MTT and LIVE/DEAD assays will be included in future investigations.

We have determined the DNA content of fresh AF and NP tissues, which was included in the Appendix A, Figure A3. In fresh AF and NP samples, the DNA content was about 89±15 and 37±10 µg per g of IVD tissue wet weight, respectively. This data is in agreement with the literature (Bezci et al., JOR Spine, 2019; Fiordalisi et al., Biomater Adv, 2022). After 21 days, we have observed a significant decrease in the DNA content of the AF (0.3±0.4-fold) and NP (0.5±0.3-fold) compared to freshy isolated tissues. Please, see reply to the previous question for possible reasons for this decrease. This was addressed in the discussion section, lines 262-266.

Reviewer 6 Report

This is a very interesting paper that discussed how Oestrogen and vibration improve intervertebral disc cell viability and decrease catabolism in bovine organ cultures. Overall, the paper is well written, and the results are solid, and the conclusion is reasonable.

Major Comments:

1.    Could the author provide a diagram to show the signal pathway that may involve in improved anabolic response by IVD cells?

2.    In the Statistical analysis, could the authors specify why nonparametric comparison test, and box plot was used for data?

Minor comments:

1.    In the Abstract, 17β-oestradiol (E2) should appear first in the text.

2.    In the Discussion part, could the authors discuss the effects of different dose of E2 and different settings of LMHFV might have on the IVD.

3.    In the Discussion part, could the authors also discuss how the effects of E2 and LMHFV might interact with each other in the IVD?

Author Response

Point-by-point response to Reviewer 6

This is a very interesting paper that discussed how Oestrogen and vibration improve intervertebral disc cell viability and decrease catabolism in bovine organ cultures. Overall, the paper is well written, and the results are solid, and the conclusion is reasonable.

Authors: We would like to thank the reviewer for the assessments. We hope to have adequately addressed all comments and recommendations. The revised version of the manuscript is submitted in attachment. Changes are highlighted in bold and red.

Major Comments:

  1. Could the author provide a diagram to show the signal pathway that may involve in improved anabolic response by IVD cells?

Authors: The authors have included in the discussion section a diagram summarizing the changes in gene expression of AF and NP cells (Figure 9, lines 386-389).

  1. In the Statistical analysis, could the authors specify why nonparametric comparison test, and box plot was used for data?

Authors: We have assessed whether the data was normally distributed using the Shapiro-Wilk test. For parametric data, the comparison between groups was performed using Brown-Forsythe and Welch one-way analysis of variance, followed by Dunnett’s multiple comparison test. For nonparametric data, the comparison between groups was performed using Kruskal-Wallis test with Dunn’s multiple comparison test. We opted to depict the data using box plots to better graphically demonstrate the range and variability of the data. The text was rephrased in the Statistical analysis section (lines 516-520).  

Minor comments:

  1. In the Abstract, 17β-oestradiol (E2) should appear first in the text.

Authors: We have corrected it in line 17.

  1. In the Discussion part, could the authors discuss the effects of different dose of E2 and different settings of LMHFV might have on the IVD.

Authors: The authors have rephrased the paragraphs discussing the effects of E2 (lines 285-321) and of LMHFV (lines 322-349) on the IVD. More literature and details on the used E2 concentrations and LMHFV settings were provided.

  1. In the Discussion part, could the authors also discuss how the effects of E2 and LMHFV might interact with each other in the IVD?

Authors: We have rephrased the discussion section (lines 350-374) to address the reviewer’s suggestion. 

Round 2

Reviewer 1 Report

The previous comments have been satisfactorily addressed by the authors.

Author Response

The authors thank the reviewer’s comments and thorough review process.

Reviewer 3 Report

The authors have made serious modifications, and we think this version is suitable for publication.

Author Response

(The authors gave the same response as above.)
